# Diagnostic and Prognostic Potential of Exosomal Cytokines IL-6 and IL-10 in Polytrauma Patients

**DOI:** 10.3390/ijms241411830

**Published:** 2023-07-23

**Authors:** Birte Weber, Ramona Sturm, Dirk Henrich, Ludmila Lupu, Katrin Rottluff, Ingo Marzi, Liudmila Leppik

**Affiliations:** 1Department of Trauma-, Hand- and Reconstructive Surgery, University Hospital Frankfurt, Goethe-University, 60596 Frankfurt am Main, Germany; ramona.sturm@kgu.de (R.S.); d.henrich@trauma.uni-frankfurt.de (D.H.); katrin.rottluff@kgu.de (K.R.); marzi@trauma.uni-frankfurt.de (I.M.); liudmila.leppik@kgu.de (L.L.); 2Institute of Clinical and Experimental Trauma-Immunology, University Hospital of Ulm, 89081 Ulm, Germany; ludmila.lupu@uni-ulm.de

**Keywords:** polytrauma, extracellular vesicle, exosomes, size exclusion chromatography, exosomal cytokines, interleukin 6, interleukin 10

## Abstract

Trauma remains a leading cause of morbidity and mortality. Polytraumatized patients need a precise, early diagnosis to avoid complications such as multiorgan failure or sepsis. Inflammatory cytokines, commonly used for diagnosis, have a short half-life, which limits their efficacy as a diagnostic or prognostic marker. In this study, we hypothesized that cytokines in exosomes could have a longer half-life, and therefore could be used as diagnostic and prognostic markers in polytrauma patients. Plasma samples from polytraumatized patients (ISS ≥ 16, *n* = 18) were collected in the emergency room (ER) 1, 2, 3 and 5 days after trauma. Plasma-exosomes were isolated via size exclusion chromatography from polytraumatized patients and healthy volunteers (*n* = 10). The systemic and exosomal concentrations of interleukin (IL)-6, IL-10, IL-1β and TNF were measured using high-sensitive ELISAs. To investigate the diagnostic and prognostic potential of exosomal cytokines, data were correlated with clinical outcome parameters (injury severity, ventilation time, time in ICU and survival) documented in the patients’ electronic records. Despite the use of high-sensitive ELISAs, IL-1β and TNF alpha were not detected in exosomes. IL-6 and IL-10 were detectable in polytraumatized patient exosomes at all time points. A decrease over time of both systemic and exosomal IL-6 concentrations was observed. Furthermore, exosomal and systemic IL-6 concentrations moderately correlated (r = 0.63). Exosomal IL-6 in the ER moderately correlated with the Injury Severity Score (ISS) (mean 35.5 ± 11.5) (r = 0.45) and was associated with non-survival in polytrauma patients (*p* < 0.05). In contrast to IL-6, no correlation between systemic and exosomal IL-10 concentrations was found. Exosomal IL-10 concentrations remained unchanged throughout the observation time, whereas systemic IL-10 concentrations peaked in the ER and were significantly reduced after 24 h. Data from this study support our hypothesis that some cytokines (IL-10), but not all (IL-6), are detectable in exosomes significantly longer than they are in plasma. This might indicate that they are protected from degradation. Although we did not find a correlation between IL-10 exosomal concentration and patient outcome, our data confirm that exosomal cytokines are of interest as potential diagnostic and prognostic markers in polytrauma patients, and require further detailed research.

## 1. Introduction

World-wide, trauma remains one of the leading causes of morbidity and mortality [1]. Polytrauma is complex and involves multiple body regions and organs being injured at the same time, causing hemorrhage and resulting shock, as well as prolonged systemic inflammation [2,3,4]. Next to initial death caused by dramatic injuries, death due to post-traumatic complications is a common problem. Especially, infectious complications leading to sepsis or multiple organ failure (MOF) drastically influence outcomes of trauma patients [5,6]. Therefore, the intense treatment of those patients represents a high economic burden for society [7]. The patients in this group, in general, need to be diagnosed as soon as possible through a combination of laboratory markers as well as radiological diagnostics in order to be able to receive prompt and accurate treatment. Next to other laboratory markers, systemic cytokines, such as IL-6, are commonly used to predict patients’ outcomes and to forecast the appearance of inflammatory complications. Systemic cytokines, such as IL-6 and IL-10, as well as the ratio of IL-6/IL-10, have been reported to be elevated proportionally to multiorgan dysfunction and mortality [8,9]. As a routine marker in intensive care units (ICUs), systemic concentrations of IL-6 predict the development of MODS with an overall accuracy of almost 85% [10]. Unfortunately, the short half-life of most inflammatory cytokines complicates their use in systemic monitoring using plasma or serum, and therefore hampers their prognostic potential. For example, TNF, one of the first cytokines released after polytrauma, peaks 2 h after trauma, and has an extremely short half-life of 20 min in serum samples [11]. IL-1β has a longer half-life of 90 min, and is responsible for the induction of cytokines and chemokines from immune cells, as well as the C-reactive protein (CRP), which is highly relevant in the clinical setting [11,12,13]. Although the half-life of IL-1β is slightly longer, it is still just a small time-window in which IL-1β can be used to monitor a patient’s outcome in the setting of emergency care. TNF-α and IL-1β boost the production of IL-6 by macrophages, while IL-6 stimulates natural killer cells, restrains apoptosis of neutrophils, and releases acute phase proteins from hepatocytes [11,12,14]. The half-life of IL-6 is 15 h, which makes it more suitable as a monitoring parameter in intensive care settings [12]. Next to IL-6, the anti-inflammatory cytokine IL-10 was also described as a prognostic marker for polytrauma. A significant correlation was found between the ISS and levels of both IL-6 and IL-10 over 24 h after trauma, and significantly increased serum concentrations of IL-6 and IL-10 were shown in patients who did not survive 30 days [15]. In patients who developed sepsis after trauma, a reduced amount of systemic IL-10 was measured [16]. The half-life of IL-10 is between 2–5 h [17]. All in all, cytokines could be helpful tools in predicting post-traumatic complications such as MOF or sepsis, but the short half-life of most cytokines drastically limits their prognostic potential. A promising alternative tool for diagnosing trauma severity and complications might be found in the broad field of exosomes [18,19]. Exosomes are defined as the smallest (150–300 nm in size) population of extracellular vesicles (EVs), and are released from healthy cells through the endosomal route [20,21,22]. Exosomes could be responsible for the transport of different proteins, nucleic acids, lipids and other small molecules, while defending them from degradation processes [23]. In this study, we hypothesized that cytokines inside exosomes could be protected from degradation, and could therefore be a more stable monitoring parameter for polytraumatized patients than systemic cytokines. To evaluate the prognostic and diagnostic potential of exosomal cytokines, we measured exosomal concentrations of IL-6, IL-10, TNF and IL-1β in polytrauma patients over the course of 5 days after trauma, compared them with systemic concentrations, and associated them with clinical outcome parameters.

## 2. Results

This analysis included 18 polytraumatized patients with an ISS ≥16 (mean ISS 35.5 ± 11.5) and 10 healthy volunteers. Exosomal and systemic cytokine concentrations of IL-6, IL-10, IL-1β and TNF were measured using ELISA and compared across groups (Table 1).

The size and concentration of isolated exosomes were characterised using NTA analysis (Figure 1A,B). No differences in the exosome size distribution or concentration among polytrauma patients’ and healthy controls’ samples were found (Appendix A). Exosomal IL-6 concentration was shown to be significantly increased in polytrauma patients (ER, day 1 and 2 samples) compared to healthy group exosomes. The maximum amount of exosomal IL-6 was detected in samples collected 1 day after trauma (Figure 1C). In contrast to IL-6, no changes in the exosomal concentration of IL-10 were found among healthy or polytrauma patient groups at any time point (Figure 1D). The exosomal concentrations of IL-1β and TNF were found to be below the detection limit of high-sensitive ELISAs.

Regarding the systemic concentration of cytokines, a significant increase in both IL-6 and IL-10 concentrations (Figure 2) was found in polytrauma patients when compared to healthy volunteers (Table 1). The maximum amount of both cytokines was detected in samples collected in the emergency room (Figure 2A,B). The systemic concentration of IL-10 significantly decreased on day 1 after trauma, whereas the systemic concentration of IL-6 decreased later, starting from day 3 after trauma. The amount of IL-1β was found to be significantly low in patients’ samples (Figure 2C,D). TNF systemic concentration was also found to be low in patients’ samples. However, a significant increase was found in samples collected on days 1, 2 and 3 after trauma.

In addition, we performed correlation analysis among exosomal and systemic cytokines concentrations, and found that although exosomal and systemic IL-6 moderately correlated (Figure 2E), exosomal and systemic IL-10 concentrations did not correlate (Figure 2F).

In order to analyze the diagnostic and prognostic potential of exosomal IL-6 and IL-10 concentrations, a correlation analysis with clinical outcome parameters (Table 2) was conducted.

We did not observe any correlation between the exosomal IL-10 concentration and any of the outcome parameters (e.g., time in hospital or ventilation time). Exosomal IL-6 measured in the emergency room (r = 0.45) and at day 1 (r = 0.32) moderately correlated with the ISS in the polytrauma group (Figure 3A,D). Furthermore, exosomal IL-6 in the ER weakly correlated, and at day 1 moderately correlated, with lactate measurements and with time in hospital (Figure 3B–F). No link between exosomal IL-6 concentrations in the ER (as well as at day 1) and catecholamines was found. In non-survivors, exosomal IL-6 measured in the ER was significantly increased, as compared to survivors, while no association was observed at day 1.

## 3. Discussion

Systemic cytokines have been commonly used in trauma patients to predict injury severity and the appearance of inflammatory complications such as sepsis or MODS. However, the short half-life of these inflammatory cytokines complicates their systemic monitoring in plasma or serum, and therefore decreases their prognostic potential. It has been found that cytokines can be released in soluble and EV-associated forms depending on the biological system. Furthermore, some cell-produced cytokines are present predominantly in EV-associated form (e.g., T cells and monocytes-produced cytokines [24]). Moreover, it was shown that the proportion of circulating and EV-encapsulated cytokines in animal models (rats) changes in reaction to an injury. As soon as 1 h after trauma, the exosomal concentration of the majority of cytokines increased. This shift correlated with the severity of trauma [25]. For instance, 1 h after blast injury, levels of exosomal cytokines such as IL-6 or TNF-α were enhanced, while systemic levels remained undetectably low. Rats with post-traumatic organ damage had the earliest rise and most pronounced concentrations of IL-1β, IL-10, TNF-α and IL-6 in exosomes from serum samples [25]. Cytokines, enclosed in the bilayer of exosomes, were expected to be biologically active upon interacting with cells [24]. All in all, the fact that conventional ELISA measurements only detect circulating, but not exosomal, cytokines indicates that the real inflammatory immune response after trauma might be underestimated.

Although data about systemic concentrations of cytokines in polytrauma patients are plentiful, little is known about concentrations of different cytokines in plasma exosomes of healthy and polytrauma individuals, and how these concentrations change over time. To evaluate the prognostic and diagnostic potential of exosomal cytokines, we measured concentrations of IL-6, IL-10, IL-1β and TNF in exosomes and systemically in polytrauma patients over 5 days, and associated these data with clinical outcome parameters. First of all, we recognized that cytokines IL-6 and IL-10 could be detected in exosomes using ultrasensitive ELISAs, while exosomal concentrations of TNF and IL-1β were below the detection limit of this method. Based on this observation, future studies need to evaluate other methods to quantify these cytokines in exosomes.

Interestingly, both exosomal IL-6 and IL-10 concentrations showed significantly different kinetics during polytrauma. Whereas IL-6 was absent in the exosomes of healthy individuals, appeared first immediately after trauma, peaked at day 1 after trauma, and decreased during next days (resembling the kinetics of systemic IL-6), there was no difference in exosomal IL-10 concentrations among healthy volunteers and polytrauma patients, regardless of the time point after trauma. Accordingly, exosomal IL-6 concentrations moderately correlated with systemic IL-6 concentrations, while exosomal IL-10 concentrations did not correlate with systemic concentrations. We speculated that the exosomes carrying these two distinctly different cytokines also play completely different roles in the mechanism of the body’s homeostasis and response to trauma, which might explain these differences in kinetics. Exosomal pro-inflammatory cytokine IL-6 seemed to reflect the systemic IL-6, while the anti-inflammatory cytokine IL-10 seemed to be constantly maintained in exosomes. Future analysis is necessary to clarify the role of exosomal cytokines in the pathomechanism of trauma.

In addition to investigating the detection of exosomal cytokines, this study also focused on their diagnostic and prognostic potential. In our analysis, exosomal IL-6 moderately correlated with injury severity, and was significantly increased in non-survivors; these data reproduce findings about systemic IL-6 concentrations after polytrauma described in the literature [8,9]. Based on the present findings, and the fact that our analysis of exosomal IL-6 was more elaborate, exosomal IL-6 as a potential biomarker appeared to provide no additional information for monitoring polytrauma patients in the first days after trauma.

Animal studies using sepsis models have shown that exosomal concentrations of some pro-inflammatory cytokines, such as IL-1β, IL-6 and TNF, increase in the early phase of sepsis, while exosomal concentration of the anti-inflammatory cytokines (IL-4 and IL-10) peak first during the late phase of sepsis [26]. In this study, none of the included patients developed sepsis during their hospital stay; therefore, our data could not be correlated with these findings. Future studies including a group of polytrauma patients with sepsis should analyse whether the development of sepsis is reflected in exosomal cytokines, and if exosomal IL-10, which was found to be constant in our study, increases in these patients.

Although little is known about exosomal cytokine concentrations in polytrauma patients, studies focused on mono-trauma (for example, on traumatic brain injuries (TBI)) demonstrated that exosomal cytokines reflect a patient’s outcome after trauma, and might be used as biomarkers for concussion, for example. It was shown that EV-associated IL-6 was significantly elevated following concussion in football players, and that this EV-associated IL-6 level was associated with the number of days of reported symptoms [27]. Gill et al. (2018) reported that IL-10 concentration was significantly increased in neuron-derived exosomes in patients with mild TBI, and was associated with post-traumatic stress-disorder [28]. Other authors reported that IL-6 concentration in CNS-enriched exosomes was increased in patients with cognitive impairment, as compared to healthy controls [29]. Interestingly, IL-6 was shown to be highly (16-fold) increased in neuron-derived exosomes in chronic mild TBI as compared to acute mild TBI [30]. It is also important to take into account that exosomes are only one sub-population of extracellular vesicles, and that bigger EVs (microvesicles) could also transport cytokines. It was shown in a murine high-air pressure model that microvesicles, but not exosomes, elevated IL-1β as a reaction to trauma [31]. These findings could partially explain our findings of no detectable levels of IL-1β in exosomes, and suggest that other fractions of extracellular vesicles should be included in similar analyses.

In summary, this study showed that systemic and exosomal cytokine profiles differ in polytrauma patients. Whereas some cytokines (TNF and IL-1β) were not detectable in exosomes using modern ELISA techniques, other cytokines (IL-6 and IL-10) could be quantified in plasma exosomes. Moreover, changes in exosomal IL-6 concentrations with time after trauma reflected the changes of this cytokine in plasma, whereas the level of exosomal IL-10 was not affected by trauma. To our knowledge, this study is the first analysis of exosomal cytokines concentrations in polytraumatized patients over a period of 5 days.

## 4. Materials and Methods

For this analysis, we included 18 polytraumatized patients, as well as 10 healthy volunteers. The multiple injured patients were all admitted to our German Level 1 Trauma Centre between 2016 and 2020. An ISS of at least 16 was required to be included in the trauma group. Ethical approval was given by the Local Ethics Committee of the University of Frankfurt (approval ID 89/19). Analysis was performed using plasma exosomes. The plasma was acquired after admission at the emergency room (ER) and 1, 2, 3 and 5 days after trauma (15 min centrifugation at 3500 g at 4 °C). Based on the patient’s electronic record, outcome parameters (time in the hospital, ventilation time, time in ICU/IMC, blood transfusion, need for catecholamines, etc.) were documented and correlated with results from the cytokine measurements.

Isolation and characterization of exosomes: Exosome isolation was conducted using 200 µL plasma from polytraumatized patients or healthy controls using Exo-Spins, commercially available as exosome size-exclusion columns (Cell Guidance Systems, Cambridge, UK). To determine exosomes’ number and size, the exosomal protein concentration (Coomassie Plus (Bradford) Assay, Thermo Fisher Scientific, Rockford, IL, USA), as well as nanoparticle tracking analysis (NTA) (Nanosight NS500, Malvern Panalytical, Kassel, Germany), were led through.

Systemic cytokine ELISAs: Systemic (plasma) concentrations of cytokines IL-6, IL-10, IL-1β and TNF were measured in plasma samples collected in the ER and 1, 2, 3 and 5 days after trauma using Quantikine ELISAs (R&D; Human IL-6: #D6050; Human IL-10: D1000B; Human IL-1β/IL-1F2: #DLB50; Human TNF-a: #DTA00D) according to the manufacturer’s instructions.

Exosomal cytokine ELISAs: To measure exosomal concentrations of cytokines, isolated exosomes were first lysed using M-PER^®^ Mammalian Protein Extraction Reagent (Thermo Fisher Scientific, Dreieich, Germany) and HaltTM proteinase inhibitor cocktail (Thermo Fisher Scientific). Next, the exosomal cytokine concentration was detected using Quantikine high-sensitive ELISA (R&D; Human IL1β: #HSLB00D (sensitivity 0.063 pg/mL); Human IL6: #HS600C (sensitivity 0.09 pg/mL); Human TNF-a: #HSTA00E (sensitivity 0.049 pg/mL); Human IL10: #HS100C (sensitivity 0.17 pg/mL)) following the manufacturer’s protocol.

Statistical analysis: Statistical analyses were conducted using Graph Pad-Prism 9 (Dotmatic, San Diego, CA, USA). The values were presented as mean ± standard error of the mean (SEM). Whether data followed a normal distribution was tested using the Kolmogoroff–Smirnow test. Data were analyzed using the Kruskal–Wallis test followed by Dunn’s multiple comparison test. For statistical analysis of two groups, the Mann–Whitney U-test was applied. For the correlation analysis, linear correlation (r) was assessed using the Spearman test. Results were considered statistically significant when * *p* ≤ 0.05, ** *p* ≤ 0.01, *** *p* ≤ 0.001, and **** *p* ≤ 0.0001.

## 5. Conclusions

This study’s data showed that some cytokines (IL-10 and IL-6), but not others (TNF alpha and IL-1β) were detectable in plasma-exosomes for several days after polytrauma. Although the concentration of IL-10 in exosomes was constantly maintained after polytrauma, exosomal IL-6 concentration moderately correlated with systemic IL-6 concentration, and could be associated with trauma severity (ISS) and outcome parameters (e.g., survival). Although we did not find a correlation between IL-10 exosomal concentration and patient outcome, our data confirm that exosomal cytokines could be potential diagnostic and prognostic markers in polytrauma patients. Further detailed research is required to better understand their role in the pathomechanism of trauma.

## Figures and Tables

**Figure 1 ijms-24-11830-f001:**
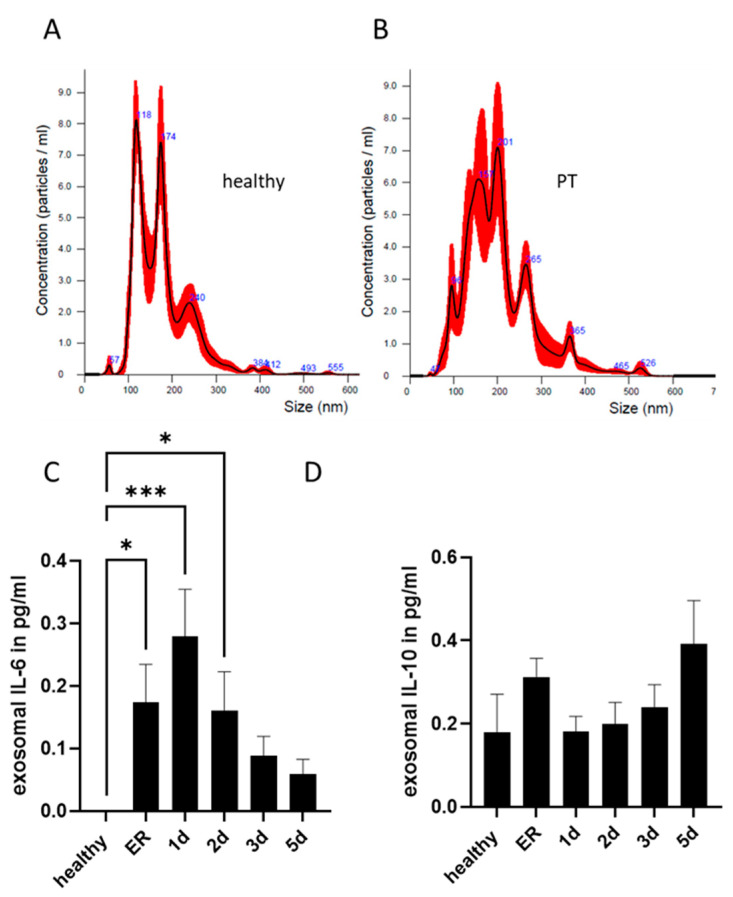
Exosomal cytokine concentrations. Representative images of NTA analysis of exosomes from (**A**) healthy controls and (**B**) polytrauma patients (PT). (**C**) Exosomal IL-6 concentration was significantly increased in polytrauma patients at emergency room (ER) 1 day (d) and 2 days after trauma as compared to healthy volunteers. (**D**) Exosomal IL-10 concentration was comparable in healthy volunteers and polytrauma patients. * *p* ≤ 0.05, *** *p* ≤ 0.001. PT *n* = 18; healthy volunteers *n* = 10.

**Figure 2 ijms-24-11830-f002:**
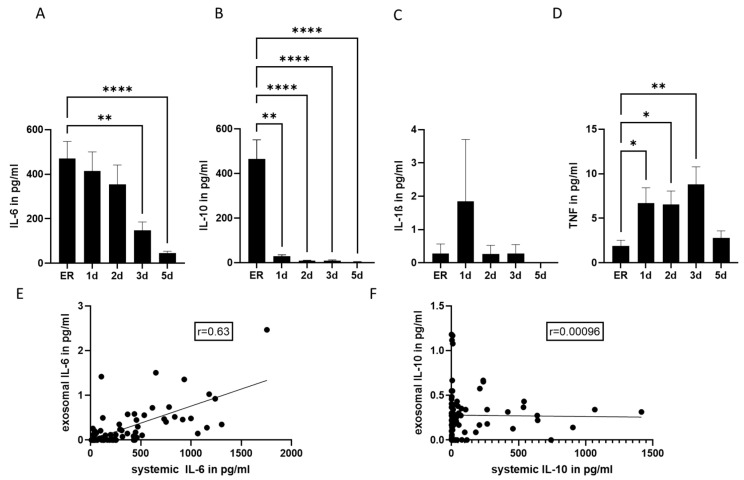
Systemic cytokine concentration. (**A**) Plasma IL-6 concentration was measured in polytrauma patients in the emergency room (ER) and 1, 2, 3 and 5 days after trauma. (**B**) Systemic IL-10 concentration in polytraumatized patients. (**C**) Systemic IL-1β and (**D**) systemic TNF in samples over 5 days after polytrauma. (**E**) Exosomal IL-6 concentration moderately correlated (r = 0.63) with systemic IL-6 concentrations. (**F**) Exosomal IL-10 did not correlate with systemic levels of IL-10. * *p* ≤ 0.05, ** *p* ≤ 0.01, **** *p* ≤ 0.0001. PT *n* = 18; healthy volunteers *n* = 10.

**Figure 3 ijms-24-11830-f003:**
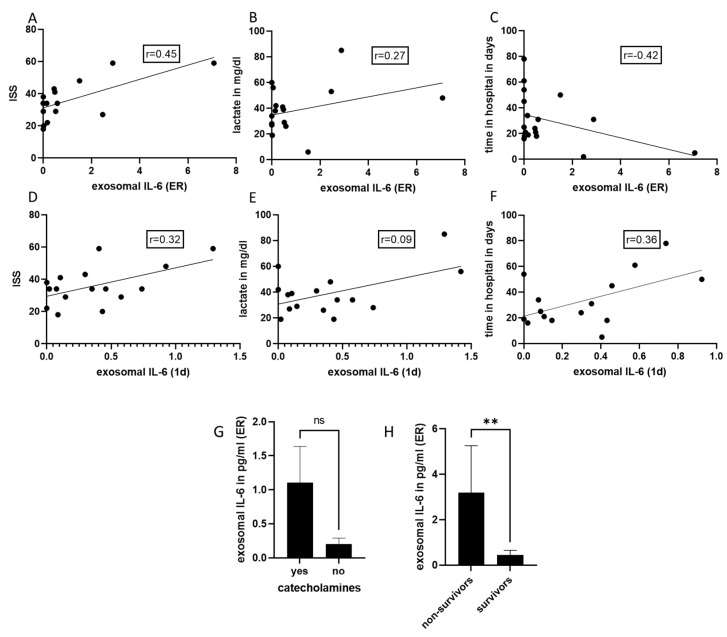
Diagnostic and prognostic potential of exosomal IL-6. (**A**) Moderate correlation between the amount of exosomal IL-6, measured in the emergency room (ER), and the Injury Severity Score (ISS). (**B**) A weak correlation was shown between exosomal IL-6 and lactate in the emergency room and (**C**) time in hospital. (**D**) Moderate correlation between exosomal IL-6 measured at day 1 after trauma, (**E**) the ISS lactate and (**F**) time in hospital. (**G**) Exosomal IL-6 in patients with and without the need for catecholamines. (**H**) Exosomal IL-6 was significantly increased in non-survivors. ** *p* ≤ 0.01; ns = non-significant. Polytrauma patients (PT) *n* = 18.

**Table 1 ijms-24-11830-t001:** Exosomal and systemic cytokines concentrations in healthy volunteers and polytrauma patients. nd = no data.

Cytokines (pg/mL)	Healthy	PT ER	PT 1d	PT 2d	PT 3d	PT 5d
Exosomal IL-6	0	0.2 ± 0.2	0.3 ± 0.3	0.2 ± 0.2	0.1 ± 0.1	0.1 ± 0.1
Exosomal IL-10	0.2 ± 0.3	0.3 ± 0.2	0.2 ± 0.1	0.2 ± 0.2	0.2 ± 0.2	0.4 ± 0.4
Exosomal Il-1β	nd	nd	nd	nd	nd	nd
Exosomal TNF	nd	nd	nd	nd	nd	nd
IL-6	0	471.0 ± 286.0	414.3 ± 332.9	354.2 ± 337.8	1057.5 ± 3115.9	45.7± 30.8
IL-10	1.5 ± 1.2	465.9 ± 352.1	28.5 ± 27.3	8.8 ± 8.4	9.8 ± 11.8	3.8 ± 4.9
Il-1β	0	0.3 ± 1.2	1.9 ± 7.4	0.3 ± 1.1	0.3 ± 1.1	nd
TNF alpha	0	1.9 ± 2.8	6.7 ± 6.6	6.6 ± 5.8	8.8 ± 7.3	2.8 ± 2.8

**Table 2 ijms-24-11830-t002:** Clinical parameters of polytrauma patients.

Clinical Parameter	Polytrauma Patients (*n* = 18)
Male/Female (%)	61/39
ISS (Mean ± SD)	35.5 ± 11.5
Death (%)	16.7
Time in ICU (days)	21.7 ± 15.6
Ventilation time (days)	15.2 ± 16.8
Time in hospital (days)	30.7 ± 19.3
Need for catecholamines (%)	77.8

## Data Availability

Not applicable.

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
