# Peer review of "Diagnostic and Prognostic Potential of Exosomal Cytokines IL-6 and IL-10 in Polytrauma Patients"

_ijms, 2023, doi:10.3390/ijms241411830_

Round 1
Reviewer 1 Report
The symbol of IL-1ß is not homogeneous in the text. it si requested to standardize the symbol.
as a suggestion is to place Table 2 before Figure 3.
Author Response
Dear Editor,
First of all, we want to thank you and the reviewers for the critical proofreading, the constructive evaluation of our manuscript and the opportunity to further improve our manuscript. Addressing the concerns in a thoroughly revised version of our manuscript critically improved its quality, hopefully meeting now the high-quality standards for publication in the International Journal of Molecular Sciences. The detailed point-to-point response to the reviewers’ comments are outlined below.
Sincerely,
Dr. med. Birte Weber
Point-by-Point Response to reviewers’ comments:
Reviewer 1:
Comment 1: The symbol of IL-1ß is not homogeneous in the text. it is requested to standardize the symbol.
Response: Thank you for this comment, we revised the IL-1β in the text.
Comment 2: as a suggestion is to place Table 2 before Figure 3.
Response: We switched the position of Table 2 and Figure 3 in the manuscript.

Reviewer 2 Report
The study addresses an emerging area in the field of cytokine biology and should be of broad interest. I have the following comments:
1. The concentrations of IL-6 found in the exosomes, although significantly different between the groups/timepoints, is far lower than the ususal activation dose (EC50). The article does not provide information about the number of exosomes between the different days the patients were sampled. I would like to see if there are changes in exosome numbers and size between the studied days. In any case, I think the cytokine values need to be normalized by the exosome numbers present.
2. I did not see any information regarding the mortality of the patients. Did all the patients survive? Can exosomal cytokines be used to determine mortality in polytrauma patients?
3. Line 103 - 104: "No differences in the exosome size distribution or concentration among polytrauma patients and healthy controls samples were found (data not shown)". This is an important piece of information that should be shown in a supplementary figure. Perhaps this is the figure in relation to comment 1 above?
The manuscript is readable and English language quality is fine.
Author Response
Dear Editor,
First of all, we want to thank you and the reviewers for the critical proofreading, the constructive evaluation of our manuscript and the opportunity to further improve our manuscript. Addressing the concerns in a thoroughly revised version of our manuscript critically improved its quality, hopefully meeting now the high-quality standards for publication in the International Journal of Molecular Sciences. The detailed point-to-point response to the reviewers’ comments are outlined below.
Sincerely,
Dr. med. Birte Weber
Point-by-Point Response to reviewers’ comments:
Reviewer 2:
Comment 1. The concentrations of IL-6 found in the exosomes, although significantly different between the groups/timepoints, is far lower than the usual activation dose (EC50).
Response: Thank you for the opportunity to clarify this point.
The exact mechanism of exosome function is still not described, however it is believed that exosomes derived from target cells and carrying encapsulated cell-specific cargos (cytokines) are selectively taken up by neighbouring or distant cells, and influence a recipient cell upon their bioactive compounds (Zhang et al. 2019). That means that in case of exosomes the cytokines function locally, and therefore their effects (concentrations) could not be compared with the effects (concentrations) of systemic cytokines.
Moreover, if we roughly recalculate the concentration of exosomal cytokines measured in our exosomal isolates into the concentration inside the exosome, it is relatively high:
Our ELISA measurements are performed in 1 ml (1x10-6 m3) of diluted exosomal lysates.
The internal volume of 1.6x107 exosomes (an average number of exosomes in our samples measured by NTA) could be calculated as:
(1.6x107) x (4/3 πR3) = (1.6x107) x 1.5 × 10−21 m3 = 2.4x10-14 m3
(Exosomes internal radius without the lipid bilayer ≈ 70 nm (Li et al. 2014))
Therefore, our exosomal cytokines are diluted 4.2x107 –fold: (1x10-6 m3): (2.4x10-14 m3) = 4.2x107 and the concentration of IL-6 inside the exosome is in the µg/ml range.
Comment 2: The article does not provide information about the number of exosomes between the different days the patients were sampled. I would like to see if there are changes in exosome numbers and size between the studied days. In any case, I think the cytokine values need to be normalized by the exosome numbers present.
Response: We thank the reviewer for this suggestion. We added information concerning the exosome particles sizes and numbers measured in our samples at different time points in Supplementary Figure 1. We did not observe any significant changes in the numbers of exosomes among different groups /time points.
Supplemental Figure 1: Nano tracking analysis of exosomes: A) Mean size of exosomes measured via NTA in healthy and polytrauma patients at different time points; B) Particles numbers measured via NTA.
As there is no significant difference in particle numbers (NTA measurements) among the groups/time points and keeping in mind, that NTA quantifications of exosomes has limitations (overestimation of the EVs number due to co-isolation of lipoproteins and EVs in biological fluids samples (Sódar et al. 2016; Vogel et al. 2021)), we prefer to show raw values and avoid normalization of cytokine concentrations to particle numbers.
Comment 3: I did not see any information regarding the mortality of the patients. Did all the patients survive? Can exosomal cytokines be used to determine mortality in polytrauma patients?
Response: Thank you for this important comment. 16.7 % of the polytraumatized patients included in this analysis died (Table 2.) As presented in figure 3, non-survivors were characterised by increased exosomal IL-6 levels.
Comment 4: Line 103 - 104: "No differences in the exosome size distribution or concentration among polytrauma patients and healthy controls samples were found (data not shown)". This is an important piece of information that should be shown in a supplementary figure. Perhaps this is the figure in relation to comment 1 above?
Response: Please refer our answer to Comment 2 of Reviewer 2.
References:
Li, Mu; Zeringer, Emily; Barta, Timothy; Schageman, Jeoffrey; Cheng, Angie; Vlassov, Alexander V. (2014): Analysis of the RNA content of the exosomes derived from blood serum and urine and its potential as biomarkers. In: Philosophical transactions of the Royal Society of London. Series B, Biological sciences 369 (1652). DOI: 10.1098/rstb.2013.0502.
Sódar, Barbara W.; Kittel, Ágnes; Pálóczi, Krisztina; Vukman, Krisztina V.; Osteikoetxea, Xabier; Szabó-Taylor, Katalin et al. (2016): Low-density lipoprotein mimics blood plasma-derived exosomes and microvesicles during isolation and detection. In: Scientific reports 6, S. 24316. DOI: 10.1038/srep24316.
Vogel, Robert; Savage, John; Muzard, Julien; Della Camera, Giacomo; Vella, Gabriele; Law, Alice et al. (2021): Measuring particle concentration of multimodal synthetic reference materials and extracellular vesicles with orthogonal techniques: Who is up to the challenge? In: Journal of extracellular vesicles 10 (3), e12052. DOI: 10.1002/jev2.12052.
Zhang, Yuan; Liu, Yunfeng; Liu, Haiying; Tang, Wai Ho (2019): Exosomes: biogenesis, biologic function and clinical potential. In: Cell & bioscience 9, S. 19. DOI: 10.1186/s13578-019-0282-2.

Round 2
Reviewer 2 Report
I would like to thank the authors for accommodating my requests for clarification/revision in the manuscript. I am satisfied with the responses and recommend the manuscript for publication.